# Structural Features and Phylogenetic Implications of 11 New Mitogenomes of Typhlocybinae (Hemiptera: Cicadellidae)

**DOI:** 10.3390/insects12080678

**Published:** 2021-07-28

**Authors:** Shuanghu Lin, Min Huang, Yalin Zhang

**Affiliations:** Key Laboratory of Plant Protection Resources and Pest Management, Ministry of Education, Entomological Museum, College of Plant Protection, Northwest A&F University, Yangling 712100, China; bosstigers@163.com (S.L.); huangmin@nwsuaf.edu.cn (M.H.)

**Keywords:** Auchenorrhyncha, leafhoppers, phylogeny, tribes

## Abstract

**Simple Summary:**

Typhlocybinae is the smallest sized and most evolved leafhopper, body length 2–4 mm, forewing with four apical cells but lacking closed preapical cells. It comprises over 6000 species (Dietrich, 2013) distributed worldwide, making it the second largest group of Cicadellidae. Previous phylogenetic analyses in this subfamily were mainly based on morphological characters and were restricted to several gene fragments. To provide further insight into the relationships of its included tribes, complete mitogenomes of two Alebrini species (*Shaddai acuminatus*, *Sobrala* sp.), two Dikraneurini species (*Dikraneura* (*Dikraneura*) *zlata*, *Robusta emeiensis*), two Empoascini species (*Alebroides salici*, *Empoasca serrata*), two Erythroneurini species (*Elbelus tripunctatus*, *Kaukania anser*), two Typhlocybini species (*Eupteryx* (*Eupteryx*) *adspersa*, *Eurhadina jarrary*), and one Zyginellini species (*Yangisunda tiani*) are newly sequenced and comparatively analyzed. The mitogenomes comprise the typical set of 37 mitochondrial genes and a large non-coding region (A+T-rich region). The acceptor arm of *trnR* is the most inconstant among all the tRNAs, due to the acceptor arm comprising unpaired bases. Phylogenetic analyses using Bayesian inference and maximum likelihood methods produced a well-resolved framework of Typhlocybinae, showed the monophyly of Typhlocybinae and its inner tribes, except for Typhlocybini and Zyginellini combined. These results provide the valuable data toward the future study of the phylogenetic relationships in this subfamily.

**Abstract:**

To explore the characteristics of mitogenomes and discuss the phylogenetic relationships and molecular evolution of the six tribes within Typhlocybinae, 11 complete mitogenomes are newly sequenced and comparatively analyzed. In all of these complete mitogenomes, the number and order of the genes are highly conserved in overall organization. The PCGs initiate with ATN/TTG/GTG and terminate with TAA/TAG/T. Almost all tRNAs are folded into the typical clover-leaf secondary structure. The control region is always variable in length and in numbers of multiple tandem repeat units. The *atp8* and *nad2* exhibits the highest evolution rate among all the PCGs. Phylogenetic analyses based on whole mitogenome sequences, with three different datasets, using both maximum likelihood and Bayesian methods, indicate the monophyly of Typhlocybinae and its inner tribes, respectively, except for Typhlocybini and Zyginellini that are paraphyletic. Finally, we confirm that Erythroneurini is a subtribe of Dikraneurini.

## 1. Introduction

Members of Typhlocybinae are in the family Cicadellidae, Auchenorrhyncha, and order Hemiptera. They are distributed worldwide and their tropical fauna, in particular, is most diverse [1,2]. Typhlocybinae is composed of six tribes (Alebrini, Dikraneurini, Empoascini, Erythroneurini, Typhlocybini and Zyginellini) in the most well-accepted taxonomy system that was proposed by Dworakowska in 1979 [3]. The monophyly of Typhlocybinae has been strongly supported by previous studies [4,5,6,7]. Typhlocybinae are the smallest sized and one of the most evolved groups of leafhoppers, body length 2–4 mm and forewing with four apical cells but lacking closed preapical cells [1,4]. It is the second largest group of Cicadellidae with over 6000 species [8]. These leafhoppers are of great ecological and economic significance. Large populations of these species feed mainly on host plant leaf parenchyma cells, causing “hopperburn” to numerous crops including apple, grape, soybean and potato [1]. In addition, some serve as vectors of numerous pathogens and viruses to host plants. Previous investigations of Typhlocybinae have focused primarily on morphology-based taxonomy and limited gene fragments or incorporated only a few taxa into higher level phylogenetic studies [3,4,5,6,7,9]. Thus, the phylogenetic relationships among tribes of Typhlocybinae has not yet been adequately explored. Further investigation based on new characteristics, including mitochondrial genomes, is used urgently to reconstruct the phylogeny of Typhlocybinae.

The insect mitogenome is typically double-strand circular DNA molecules, containing 13 protein coding genes, 2 ribosomal RNA, 22 transfer RNA genes and a control region (A + T-rich region) of variable length and number of multiple tandem repeat units, that regulate transcription and replication [10,11,12]. The mitogenome provides genome-level information, including base composition, sequence arrangement, codon usage or variation, RNA secondary structures and control region characters. The mitogenome has unique features including matrilineal inheritance, intron deletion, low recombination and high evolutionary rates [12,13,14]. With the recent cost reduction of high-throughput sequencing, partial or complete mitogenomes are extensively used in population genetics, phylogeny, evolution and phylogeography investigations of insects during these past several years [11,15,16,17,18,19,20,21,22].

Previously, there were only 14 complete mitogenomes pertaining to four tribes (Empoascini, Erythroneurini, Typhlocybini and Zyginellini) of Typhlocybinae in GenBank (https://www.ncbi.nlm.nih.gov, accessed on 12 April 2021). However, mitogenomes of Alebrini and Dikraneurini species were still absent. In this study, we use three different datasets (PCG123, PCG123R, PCG12) and use both maximum likelihood (ML) and Bayesian inference (BI) methods to reconstruct their phylogenetic relationships and provide further insight into their taxonomic status. This is based on 11 newly sequenced and functional annotated complete mitogenomes of two Alebrini species (*Shaddai* sp., *Sobrala* sp.), two Dikraneurini species, (*Dikraneura* (*Dikraneura*) *zlata*, Dikraneurini sp.), two Empoascini species (*Alebroides salicis*, *Empoasca serrata*), two Erythroneurini species (*Elbelus tripunctatus*, *Kaukania anser*), two Typhlocybini species (*Eupteryx* (*Eupteryx*) *adspersa*, *Eurhadina jarrary*), and one Zyginellini species (*Yangisunda tiani*) along with 13 previously available mitogenomes in GenBank. The genomic size, base composition and skewness, sequence arrangements, codon usage or variation, evolutionary rate, genetic distance, ka/ks, start and stop codons, RNA secondary structures and control region characters are analyzed.

## 2. Materials and Methods

### 2.1. Sample Preparation and Genomic DNA Extraction

Collection information for the specimens sequenced in this study is shown in Appendix A. All sample species are widely distributed in China and their status are stable in the taxonomy system that was proposed by Dworakowska in 1979 [3], all the specimens were adults captured alive and preserved in 100% ethanol and stored in a −20 °C freezer in the Entomological Museum of Northwest A&F University (NWAFU) until identification and DNA extraction. Specimens were identified to species according to their wing venations and male genitalia characteristics. Genomic DNA was extracted from the whole body of adult specimens using the EasyPureR Genomic DNA Kit following the manufacturer’s protocol (TransGen, Beijing, China).

### 2.2. Sequence Assembly, Annotation and Bioinformatic Analysis

Complete mitogenomes of all 11 species were sequenced using next-generation sequencing at the Illumina HiSeq 2500 platform with paired reads of 2 × 150 bp by the Biomarker Technologies Corporation (Beijing, China), except for *Shaddai* sp. and *Sobrala* sp. that were sequenced by Novogene Corporation (Beijing, China). The raw paired-end clean reads were quality-trimmed and assembled by Geneious 8.1.3 (Biomatters, Auckland, New Zealand) with default parameters, *Empoasca onukii* (MG190360), *Empoascanara dwalata* (MT350235) and *Limassolla lingchuanensis* (NC_046037) were utilized as references. All 11 complete mitogenomes are annotated in Geneious 8.1.3 (Biomatters, Auckland, New Zealand) [23]. 13 PCGs were predicted as open reading frames (ORFs), employing the invertebrate mitochondrial genetic code. Two rRNA genes and control regions were identified by the boundary of their adjacent genes and based on comparison with homologous genes from other Typhlocybinae species. The locations and secondary structures of 22 tRNA genes were predicted by the MITOS Web Server (http://mitos2.bioinf.uni-leipzig.de/index.py, accessed on 12 April 2021) [24]; their secondary structures were then plotted manually using Adobe Illustrator CC2018 according to the predictions. The control region (A+T-rich region) tandem repeats were identified by the tandem repeats finder online server (http://tandem.bu.edu/trf/trf.html, accessed on 12 April 2021) [25]. The mitogenome maps were produced by the CGView Server (http://stothard.afns.ualberta.ca/cgview_server/, accessed on 12 April 2021) [26]. Analyses of the 11 complete mitogenomes, including nucleotide composition, composition skew, and relative synonymous codon usage (RSCU) were analyzed with PhyloSuite v1.2.2 [27]. Strand asymmetry was computed according to the formulas AT-skew = (A − T)/(A + T) and GC-skew = (G − C)/(G + C) [28]. To detect the nucleotide diversity (Pi) of 13 PCGs among 22 complete mitogenomes of Typhlocybinae, a sliding window analysis using 200 bp and a step size of 20 bp were conducted by the DnaSP v6 [29]. The ratios of non-synonymous (Ka) and synonymous (Ks) substitutions rates for each PCG of 22 Typhlocybinae species were also computed using DnaSP v6. Genetic distances between 22 Typhlocybinae species based on each PCG were computed using MEGA X with Kimura-2-parameter [30].

### 2.3. Sequence Alignment and Phylogenetic Analysis

One Evacanthinae species (*Evacanthus heimianus*) [31] and one Coelidiinae species (*Taharana fasciana*) [32] were available and chosen as outgroups. A total of 22 complete mitogenomes (including 11 newly-sequenced and 11 available mitogenomes [33,34,35,36,37,38,39,40,41]) of Typhlocybinae species (including 2 Alebrini species, 4 Empoascini species, 2 Dikraneurini species, 5 Erythroneurini species, 5 Typhlocybini species and 4 Zyginellini species) were selected to perform the phylogenetic analysis (Table 1). All available mitogenomes in this study were obtained from GenBank.

PCGs, rRNAs and tRNAs were extracted in PhyloSuite v 1.2.2. PCGs were aligned using the MAFFT v 7.313 plugin in PhyloSuite v 1.2.2, rRNAs were aligned using MAFFT version 7 online server (https://mafft.cbrc.jp/alignment/server/) (accessed on 12 April 2021) with the Q-INS-i strategy [42] Gblocks v 0.91b [43] plugin in PhyloSuite v 1.2.2 was used to remove gaps and ambiguous sites. Aligned PCGs and rRNAs were concatenated using PhyloSuite v 1.2.2, respectively.

Phylogenetic relationships were reconstructed by the maximum likelihood (ML) and the Bayesian inference (BI) methods based on three datasets: (1) the PCG123 matrix (comprising all codon positions of the 13 PCGs, 10,860 bp in total); (2) the PCG123R matrix (comprising all codon positions of the 13 PCGs and 2 rRNA, 12,998 bp in total); (3) the PCG12 matrix (comprising the first and second codon positions of the 13 PCGs, 7240 bp in total). The best partitioning schemes and evolution models of both ML analyses and BI analyses were inferred using PartitionFinder v 2.1.1 plugin in PhyloSuite v 1.2.2 using the greedy search algorithm with branch lengths linked and Bayesian information criterion (BIC) [44]. The optimal substitution results are shown in Appendix A. ML analyses were conducted using IQ-TREE v 1.6.8 plugin in PhyloSuite v 1.2.2 under ultrafast bootstraps with 1000 replicates [45]. BI analyses were conducted using MrBayes v 3.2.6, executed in the CIPRES Science Gateway (www.phylo.org) (accessed on 13 April 2021), with default settings and 5 × 106 million Markov chain Monte Carlo (MCMC) generations and sampling every 1000 generations [46]. When the average standard deviation of split frequencies fell below 0.01, it was considered to reach stationarity. The initial 25% of sampled data were cast off as burn-in, and the remaining trees were applied to generate a consensus tree and compute the posterior probabilities (PP).

## 3. Results and Discussion

### 3.1. Mitogenome Structure and Nucleotide Composition

Structures and lengths of the 11 newly sequenced complete and circularized mitogenomes of Typhlocybinae in this study are shown in Figure 1 and Appendix A. The length of mitogenomes are range from 15,131 to 17,575 bp, the length difference mainly due to the variable length of the A+T-rich region [47]. The number and order of the genes are highly conserved in overall organization, containing 13 PCGs, 22 tRNAs, 2 rRNAs and a control region. Among the 37 mitochondrial genes, 9 PCGs (*nad2, cox1, cox2, atp8, atp6, cox3, nad3, nad6, cytb*) and 14 tRNAs (*trnI, trnM, trnW, trnL2, trnK, trnD, trnG, trnA, trnR, trnN, trnS1, trnE, trnT, trnS2*) are transcribed from the majority strand (J-strand), while the remaining genes, 4 PCGs (*nad5, nad4, nad4L, nad1*), 8 tRNAs (*trnQ, trnC, trnY, trnF, trnH, trnP, trnL1, trnV*) and 2 rRNAs (*rrnL, rrnS*) are encoded on the minority strand (N-strand).

The 11 mitogenomes present a heavy AT nucleotide bias, with AT% across the whole sequence attaining 74.8–78.4%, similar to other previously sequenced mitogenomes of Typhlocybinae species [33,34,35,36,37,38,39,40,41]. The composition skew analysis shows that all 11 mitogenomes present a positive AT-skew and a negative GC-skew in the whole mitogenome. However, in tRNAs, rRNAs and PCGs with its 1st codon and 2nd codon position, it presents an opposite result. In PCGs, the 2nd codon position has the lowest AT content, while the 3rd codon position has the highest AT content. The AT content of tRNAs is lower than in rRNAs. The 11 control regions show a variable AT content, from 59.3% to 93.8% (Appendix A).

### 3.2. Protein-Coding Genes and Codon Usage

The total lengths of the 13 PCGs among the 11 complete mitogenomes range from 10,854 (*Shaddai* sp.) to 10,962 bp (*Dikraneurini* sp.). The *atp8* is the smallest gene while the *nad5* is the largest gene. The total 13 PCGs also present a heavy AT nucleotide bias, with AT% ranging from 74.1% to 76.5%, but with a negative AT-skew and a positive GC-skew. Most of the PCGs start with ATN, but *atp8*, *nad5* and *cox2* start with TTG that occurred in some mitogenomes within the 11 species. GTG was found to be the initiating codon of *cox2* in *Empoasca serrata*. The latter two initiating codons were also found in the previously sequenced mitogenomes of Cicadellidae. The PCGs always terminated with TAA, while the incomplete stop codon T appeared 33 times among all PCGs of the 11 complete mitogenomes, TAG was used the least, only 15 times (Appendix A). The incomplete stop codon T may be changed into TAA by post-transcriptional polyadenylation during the mRNA maturation process [48].

The relative synonymous codon usage (RSCU) of the 22 complete mitogenomes of Typhlocybinae used in this study, was computed and is presented in Figure 2, Appendix A, showing a roughly similar RSCU among all 22 species. The most frequently used codons are AUU-Ile, UUA-Leu2, UUU-Phe and AUA-Met, all of which they are composed wholly of A and U, indicating the codon usage has a heavy AT nucleotide bias, which explains the nucleotide AT bias in the PCGs among typhlocybine species. Additionally, the codon Arg (CGC) in *Bolanusoides shaanxiensis* was not observed.

### 3.3. Transfer and Ribosomal RNA Genes

Each of the complete mitogenomes is composed of 22 tRNA genes and 2 rRNAs. The positions of all 22 tRNAs are dispersed throughout the whole mitogenomes, and the two rRNA were identified by the boundary of the adjacent genes and based on the comparison with homologous genes from other Typhlocybinae species. The total length of tRNAs among the 11 newly sequenced mitogenomes ranges from 1383 to 1455 bp, while the total length of rRNAs ranges from 1805 to 1956 bp. Both kinds of total RNAs present a negative AT-skew and positive GC-skew. Meanwhile, with a heavy AT nucleotide bias, the AT content attains 76.1–79.0% in total tRNAs, while in total rRNAs the AT content reaches 79.8–83.0%, which is higher than that in total tRNAs (Appendix A). As presented in Figure 3 and Appendix A, all tRNAs folded into the typical clover-leaf secondary structure, except for some tRNAs with a reduced arm, presenting a simple loop or composed of unpaired bases. Examples include the DHU arm of *trnS1* in *Shaddai* sp., *Sobrala* sp., Dikraneurini sp., *Yangisunda tiani* and TΨC arm of *trnQ* in *Sobrala* sp. presenting a simple loop, commonly present in the previously sequenced mitogenomes of Cicadellidae. However, *trnR* is the most inconsistent among all the tRNAs due to the acceptor arm being composed of unpaired bases, such as the acceptor arm of *trnR* in *Elbelus tripunctatus* and *Yangisunda tiani* only having 4 or 5 bp, *Dikraneura* (*D.*) *zlata* and *Eurhadina jarrary* acceptor arm of *trnR* having 3 or 6 bp, *Shaddai* sp. where the acceptor arm of *trnR* has 6 or 7 bp. These unusual features are seldomly reported in Typhlocybinae [22]. Unpaired bases are also found in the TΨC arm of *trnQ* in *Shaddai* sp. and the anticodon arm of *trnS1* in *Sobrala* sp. and Dikraneurini sp. All of the above unusual characteristics may reveal mitogenome divergences among different species [49].

From Figure 3 and Appendix A, we observed that all tRNAs of the 11 newly sequenced mitogenomes are highly conserved in lengths of 7 bp for the acceptor arm, 7 bp for the anticodon loop, and 5 bp for the anticodon arm, while the lengths of the variable loop, DHU and TΨC arms are fickle. Additionally, we recognize a total of eight types of unmatched base pairs in the arms of tRNAs from the figures (A–C, A–G, U–C, U–G, A–A, U–U, C–C, G–G), of which C–C and G–G unmatched base pairs have seldomly been reported in previous mitogenomes of Typhlocybinae. The total number of unmatched base pairs found was 47 in *Shaddai* sp., 34 in *Sobrala* sp., 30 in *Alebroides salicis*, 35 in *Empoasca serrata*, 40 in Dikraneurini sp., 35 in *Dikraneura* (*D.*) *zlata*, 47 in *Kaukania anser*, 57 in *Elbelus tripunctatus*, 48 in *Eupteryx* (*E.*) *adspersa*, 47 in *Eurhadina jarrary* and 42 in *Yangisunda tiani.*

### 3.4. Control Region

The control region, also named the A + T-rich region, is supposed to participate in regulation of transcription and replication [10,11,12]. Located at the position between *rrnS* and *trnI*, it is the longest intergenic spacer in the mitogenome, with AT content ranging from 59.3% (*Alebroides salicis*) to 93.8% (*Yangisunda tiani*) and length ranging from 920 (*Yangisunda tiani*) to 3328 bp (*Shaddai* sp.) among the 11 newly sequenced mitogenomes. The nucleotide composition of the control region presents a positive AT skew and negative GC skew in *Shaddai* sp., *Sobrala* sp., *Dikraneura* (*D.*) *zlata*, Dikraneurini sp., *Alebroides salicis*, *Elbelus tripunctatus*, *Eurhadina jarrary* and *Yangisunda tiani*, while the rest of the species present a contrary strand asymmetry (Appendix A). All of the control regions of the 11 newly sequenced mitogenomes have tandem repeats, except for *Elbelus tripunctatus*. The fragment lengths, nucleotide sequences, positions and copy numbers of repeat units are highly variable (Figure 4).

### 3.5. Gene Overlaps and Intergenic Spacers

Appendix A reveals that for the numbers ranging from 10 to 16 in each of the 11 mitogenomes, the longest overlap is 10 bp between *trnS2* and *nad1* in Dikraneurini sp., and the gene overlaps appear more frequently between tRNAs, which is of interest concerning the lesser evolutionary restraints of tRNAs [50].

We recognize 6–14 intergenic spacers in each of the 11 mitogenomes, with lengths ranging from 1 to 57 bp. The longest intergenic spacer was found in *Elbelus tripunctatus*, between *nad5* and *trnH.* The intergenic spacers can be found in most mitogenomes, presumed to be essential for the transcriptional machinery to recognize the transcription termination site [51,52].

### 3.6. Nucleotide Diversity and Evolutionary Rate Analysis

The nucleotide diversity (Pi values) based on 13 PCGs among the 22 Typhlocybinae mitogenomes was computed by the sliding window analysis. The result is exhibited in Figure 5A, showing that the Pi values of 13 PCGs range from 0.206 to 0.382, of which, *atp8* and *nad2* have relatively high Pi values of 0.382 and 0.374 indicating the highest variability. The genes *cox1* and *nad1* have relatively low Pi values of 0.206 and 0.235, indicating that they are the most conserved genes among the 13 PCGs. Pairwise genetic distance analysis was also conducted based on 13 PCGs among the 22 Typhlocybinae mitogenomes. The congruent results (Figure 5B) show genetic distances ranging from 0.24 to 0.53, of which, *atp8* and *nad2* have relatively high distances of both 0.53, indicating the fastest evolution, while *cox1* and *nad1* have relatively low distances of 0.24 and 0.28, indicating the slowest evolution. Average non-synonymous (Ka) and synonymous (Ks) substitution rates (Ka/Ks) analyses were used to estimate the evolutionary rate of each of the 13 PCGs among the 22 Typhlocybinae mitogenomes. Ka/Ks rates ranged from 0.16 to 0.84, between 0 and 1, meaning that all 13 PCGs are evolving under the purifying selection. Among the 13 PCGs, *atp8* and *nad2* have relatively high Ka/Ks rates of 0.84 and 0.62 indicates the weakest purifying pressure, while *cox1* and *cytb* have relatively low Ka/Ks rates of 0.16 and 0.24 indicates the strongest purifying pressure.

The *cox1* gene is one of the most universal markers for identifying species and analyzing the phylogenetic relationships of leafhoppers [53,54,55], but it has the lowest Pi values, genetic distance and Ka/Ks rates, indicating it is the most conserved gene within the 13 PCGs among the 22 Typhlocybinae mitogenomes. Meanwhile, *atp8* and *nad2* are the fastest evolving genes, suggesting that *atp8* and *nad2* would be two ideal candidate markers for sibling species delimitation and population genetic differentiation in Typhlocybinae.

### 3.7. Phylogenetic Relationships

The phylogenetic relationships of 22 Typhlocybinae species and two outgroup species were reconstructed by the ML and BI methods under the best models, based on three datasets, PCG123, PCG123R and PCG12, and are almost congruent. Furthermore, most nodes indicate strong support (Figure 6, Appendix A). The 22 typhlocybine species used in this study formed a monophyletic group with respect to the two outgroup species. Additionally, Alebrini, Empoascini, Dikraneurini and Erythroneurini are also recovered as monophyletic clades, respectively, and relationships among the tribes are well resolved. However, all members of Typhlocybini and Zyginellini are clustered into a clade, indicating that the two groups are mutually paraphyletic. This is similar to the studies of Ahmed (1983), Dietrich (2013) and Chen (2021) to some extent, that divided Typhlocybinae into five tribes: Alebrini, Dikraneurini (=Forcipatini), Empoascini (=Jorumini), Erythroneurini and Typhlocybini (=Eupterygini, Zyginellini) but different from the studies of Dworakowska (1979) and Zhang (1990) based on morphological data that divided Typhlocybinae into six tribes (Alebrini, Dikraneurini, Empoascini, Erythroneurini, Typhlocybini and Zyginellini) [3,8,9,56,57], although this is currently the most widely held tribal classification system of Typhlocybinae. Some other tribal classification of Typhlocybinae having recognized 4–10 tribes that are too different from ours to be acceptable [58,59]. Here in our study, we support Zyginellini as a synonym of Typhlocybini, as treated or as the result revealed in phylogenetic studies based on molecular data of Typhlocybinae [4,8,22,57,60].

Inter-tribe relationships of the five monophyletic tribes are identical to the one obtained in the Anchored Hybrid Enrichment (AHE) based phylogenomics [6], with Alebrini sister to Empoascini as the basal branch, followed by the branch of Dikraneurini sister to Erythroneurini, and finally the branch with members of Typhlocybini and Zyginellini, indicating the relationships among the five monophyletic tribes are to some extent stable. It is different from the sister-group relationship of Empoascini to Dikraneurini that proposed by Zhang (1990) and Xu (2021) based on morphological data and Dikraneurini to Typhlocybini and Zyginellini proposed by Chen (2021) based on molecular data [9,57,61]. Therefore, Typhlocybinae divides into four tribes (Alebrini, Dikraneurini, Empoascini and Typhlocybini) and Erythroneurini as a subtribe of Dikraneurini as proposed by Balme based on morphological data and 16S rRNA with histone [4] are supported. To sum up, distinguishing relationships based solely upon morphological data is not reliable. Nevertheless, we did not verify that Empoascini was not to be as uniform as previously believed and Typhlocybini can be subdivided into subtribes [60,62], a broader analysis with more mitogenomes of representative samples is necessary to confirm these results and to elucidate the status of Typhlocybinae within Membracoidea.

## 4. Conclusions

In our study, 11 complete mitogenomes were newly sequenced and the phylogenetic relationships of the six tribes (Alebrini, Dikraneurini, Empoascini, Erythroneurini, Typhlocybini and Zyginellini) within Typhlocybinae were comparatively analyzed. Compared to other previously reported complete mitogenomes, all of these complete mitogenomes with number and order of the genes were highly conserved in overall organization. The PCGs initiate with ATN/TTG/GTG, and terminate with TAA/TAG/T. All tRNAs fold into the typical clover-leaf secondary structure, except for some tRNAs with a reduced arm, presenting a simple loop or composed of unpaired bases. Ka/Ks and genetic distance analyses indicate that the *atp8* and *nad2* exhibit the highest evolutionary rate among all of the PCGs. Phylogenetic analyses based on whole mitogenome sequences, with three different datasets (PCG123, PCG123R, PCG12), using both maximum likelihood and Bayesian methods, indicated the monophyly of Typhlocybinae and its inner tribes, respectively, except for Typhlocybini and Zyginellini being mutually paraphyletic. Topologies of the monophyletic tribes/subtribes showed Alebrini sister to Empoascini as the basal branch, followed by branch of Dikraneurini and Erythroneurini, and finally Typhlocybini. Our study provided the valuable data and efficient framework for the future phylogenetic research of Typhlocybinae.

## Figures and Tables

**Figure 1 insects-12-00678-f001:**
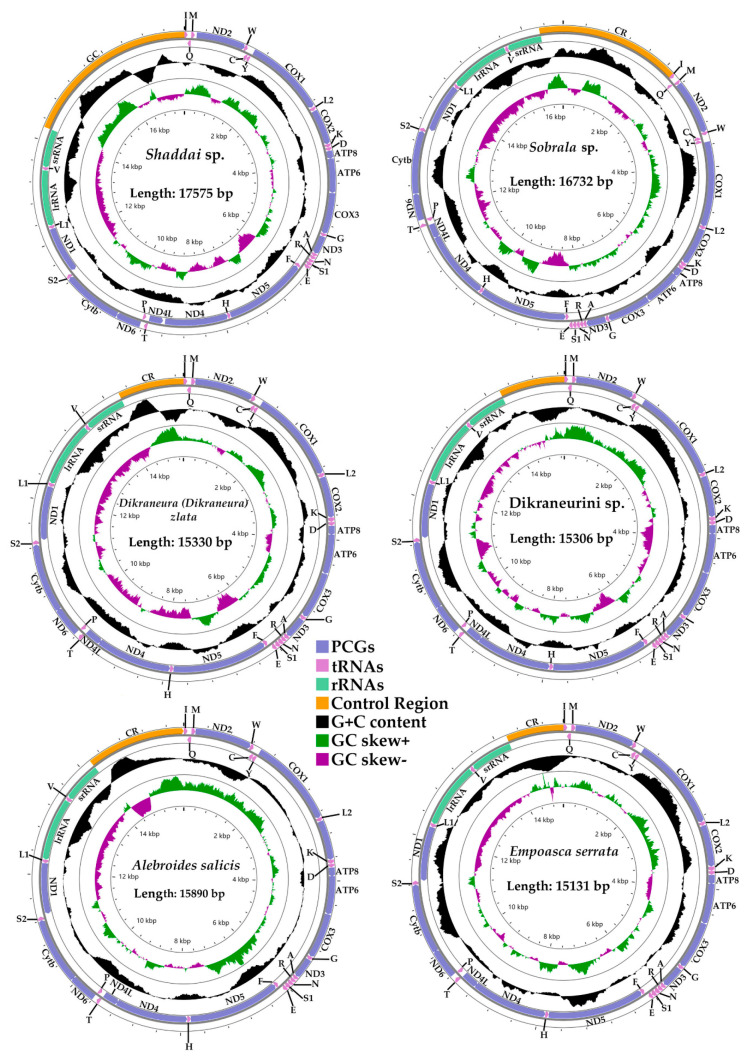
Circular map of the mitochondrial genome of *Shaddai* sp., *Sobrala* sp., *Dikraneura* (*Dikraneura*) *zlata*, *Dikraneurini* sp., *Alebroides salici* and *Empoasca serrata*.

**Figure 2 insects-12-00678-f002:**
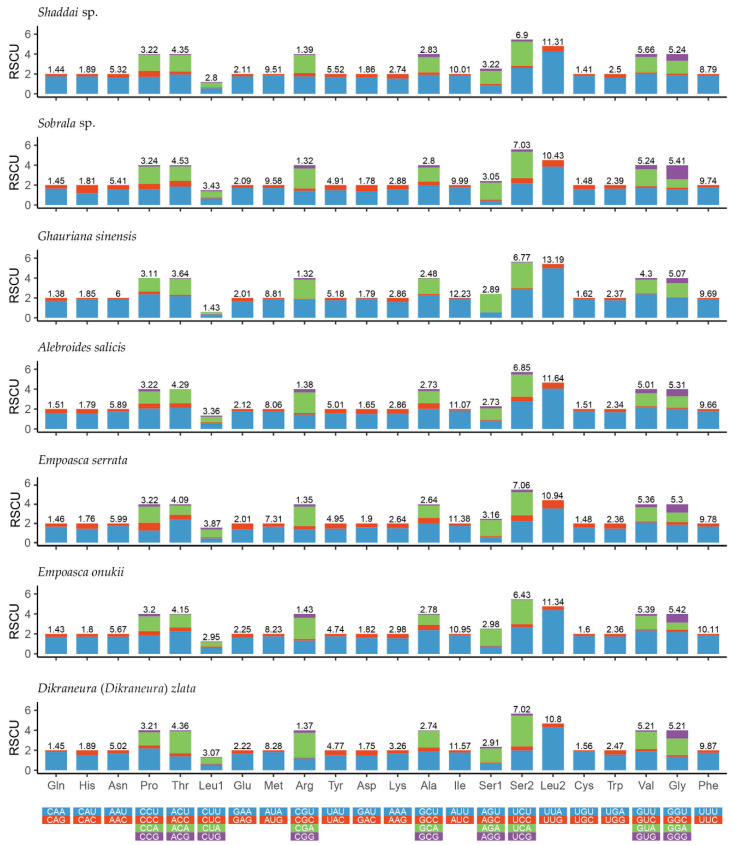
Relative synonymous codon usage (RSCU) in the mitogenomes of *Shaddai* sp., *Sobrala* sp., *Alebroides salicis*, *Empoasca serrata*, *Empoasca onukii* and *Dikraneura* (*D.*) *zlata*.

**Figure 3 insects-12-00678-f003:**
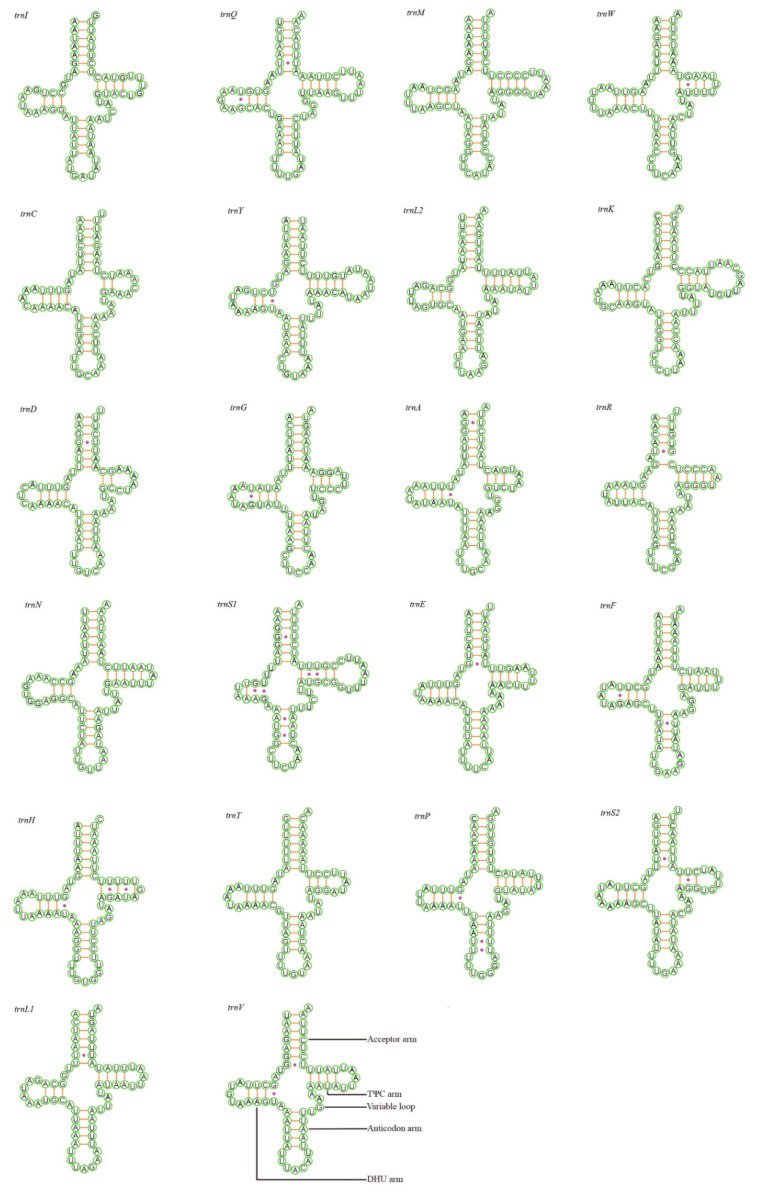
Predicted secondary cloverleaf structure for the tRNAs of *Alebroides salicis*.

**Figure 4 insects-12-00678-f004:**
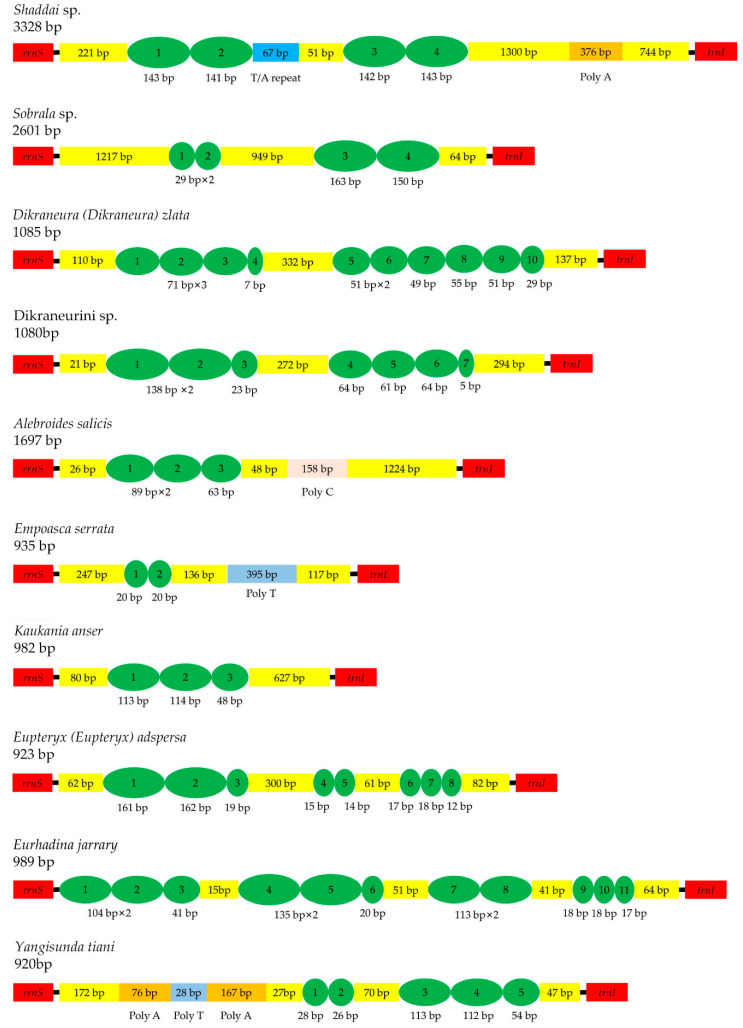
Organization of the control regions in the mitochondrial genomes of *Shaddai* sp., *Sobrala* sp., Dikraneura (Dikraneura) zlata, Dikraneurini sp., *Alebroides salicis*, *Empoasca serrata*, *Kaukania anser*, Eupteryx (Eupteryx) adspersa, *Eurhadina jarrary* and *Yangisunda tiani*. The green ovals indicate the tandem repeats; the yellow boxes indicate the non-repeat regions.

**Figure 5 insects-12-00678-f005:**
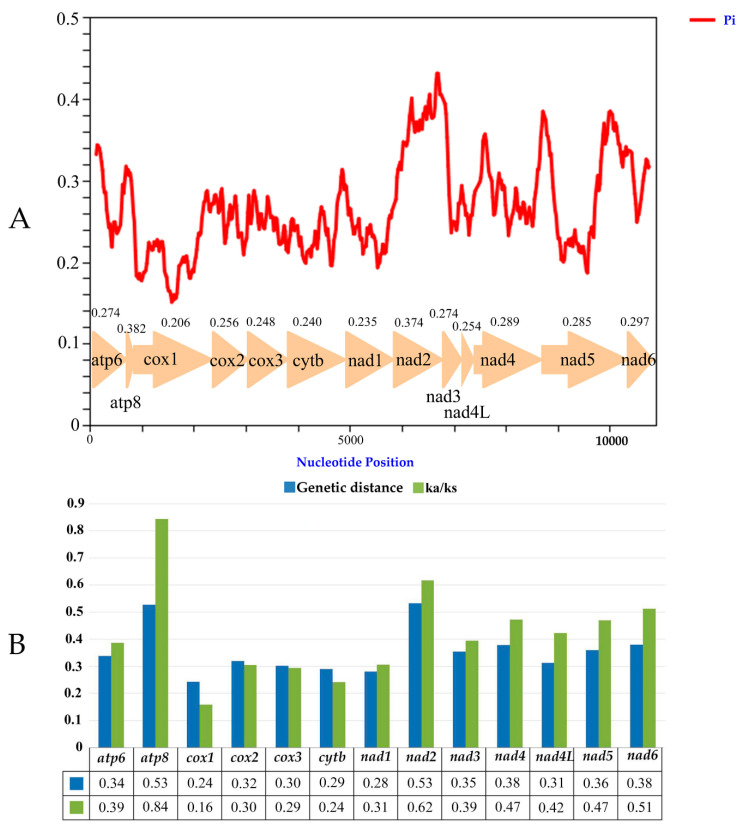
(**A**) Sliding window analyses of 13 PCGs among 22 species of Typhlocybinae, the red line shows the value of nucleotide diversity Pi. (**B**) Genetic distances and the ratio of non-synonymous (Ka) to synonymous (Ks) substitution rates of 13 PCGs among 22 species of Typhlocybinae, the average value for each PCG is shown under the gene name.

**Figure 6 insects-12-00678-f006:**
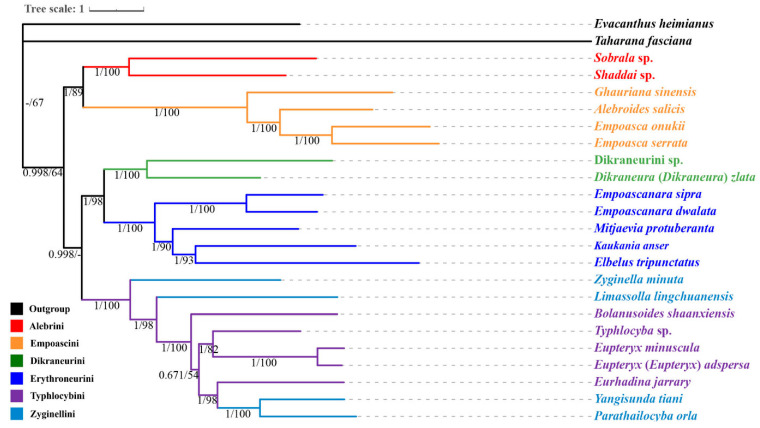
The phylogenetic tree produced using BI methods based on the dataset of PCG123. ML and BI analyses showed the same topology. The numbers under the branches are Bayesian posterior probabilities (PP) and bootstrap support values (BS).

**Table 1 insects-12-00678-t001:** Classification and origins of the mitogenomic sequences used in this study.

Family	Subfamily	Tribe	Species	Accession Number	Reference
Cicadellidae	Evacanthinae		*Evacanthus heimianus*	MG813486	[31]
	Coelidiinae		*Taharana fasciana*	NC_036015	[32]
	Typhlocybinae	Alebrini	*Shaddai* sp.	MZ014457	This study
			*Sobrala* sp.	MZ014458	This study
		Empoascini	*Ghauriana sinensis*	MN699874	[33]
			*Alebroides salicis*	MZ014449	This study
			*Empoasca serrata*	MZ014453	This study
			*Empoasca onukii*	NC_037210	[34]
		Dikraneurini	*Dikraneura* (*Dikraneura*) *zlata*	MZ014450	This study
			Dikraneurini sp.	MZ014451	This study
		Erythroneurini	*Empoascanara dwalata*	MT350235	[35]
			*Empoascanara sipra*	NC_048516	[36]
			*Mitjaevia protuberanta*	NC_047465	[37]
			*Elbelus tripunctatus*	MZ014452	This study
			*Kaukania anser*	MZ014456	This study
		Zyginellini	*Zyginella minuta*	MT488436	[38]
			*Limassolla lingchuanensis*	NC_046037	[39]
			*Parathailocyba orla*	MN894531	[40]
			*Yangisunda tiani*	MZ014459	This study
		Typhlocybini	*Bolanusoides shaanxiensis*	MN661136	Unpublished
			*Typhlocyba* sp.	KY039138	[41]
			*Eupteryx* (*Eupteryx*) *adspersa*	MZ014454	This study
			*Eupteryx minuscula*	MN910279	Unpublished
			*Eurhadina jarrary*	MZ014455	This study

## Data Availability

Data available on request.

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
