# Peer review of "Structural Features and Phylogenetic Implications of 11 New Mitogenomes of Typhlocybinae (Hemiptera: Cicadellidae)"

_insects, 2021, doi:10.3390/insects12080678_

Round 1

Reviewer 1 Report

Dr Lin and Collaborators have sequenced the mitochondrial genome of 11 species of Typhlocybinae, including representatives of all tribes. They describe the genomes in detail and provide a phylogenetic hypothesis for tribe level relationships based on the new genomes as well as others from the literature.

I think the study is well structured and has been conducted according to the good practices in the field. I am listing some aspects to reconsider in more detail before acceptance, but in the end I deem the manuscript worth of publication.

Simple summary and Abstract are a little over-technical. Please rewrite, keeping in mind that these sections are meant to convey the significance of the study in easy terms for the general reader, not necessarily interested in details that will nevertheless be dealt with in the manuscript body.

Page 2 line 74 and following: it is stated that 14 genomes were available for Typlocybinae and 11 were newly sequenced. This nevertheless is at odd with the number of genomes used in the phylogenetic analysis (22 from Typlocybinae plus 2 outgroups). To my reading, 3 genomes of Typlocybinae available in the literature were removed before phylogenetic analysis. If this is a misunderstanding, please clarify. If some genomes were excluded from the phylogenetic analysis, please justify the rationale behind this choice or, better, include all.

Methods/Results, in general: part of the interest in sequencing complete mitochondrial genomes is that they can be reused in following studies by the Author and/or other Authors. As such, it is mandatory to convince the reader that the sequences as submitted are correct from a technical standpoint beyond any doubt. I would clarify on the following:

- which is the average coverage of each genome? If they are all high, a minimum figure can be indicated for the whole set. Is there any major inequality in genome coverage that would suggest the possibility of a misassembly?

- control regions with long repeated sequences are difficult to assemble using short reads. Using a genome from the literature as a guide may foster control region completion, but also force completion even in the absence of strong evidence. Describe which actions were taken to check the correct assembly of control regions.

- please specify if each species has a different library index or if reads from all species come from the same library and have been assigned to individual species based solely on the assembly. In the latter case, please indicate which actions were taken to ensure that no cross assembly has taken place between species. Options I would recommend are to construct 13 single gene trees and ensure topologies are identical or to visualize long/almost perfect similarities across genomes (http://ezmito.unisi.it/ezmix) to highlight sequences that are present in more than one genome.

Page 3 line 94: it is stated that samples were identified to species, but this is not the case, as evident for three species (Shaddai sp., Sobralia sp. and Dikraneurini sp.) for which species information is missing. Regarding Shaddai sp. and Sobralia sp., that are identified to the genus level, the authors should justify missing species information. I'd be ready to justify lack of species information if there are known taxonomical issues in the genus or if the genera/species are under revision, not otherwise. Regarding Dikraneurini sp., identified to the level of tribe, I do not think it is acceptable to have a mitochondrial genome sequenced with such an uncertainty on the species. I suggest the authors reach the point of a determination at least to the genus level (if this can be justified, see above) or remove the sequence altogether.

Table 1: please insert horizontal lines to make it more easily readable.

Page 5 line 154: I assume it is 106 millions of generations, not 106.

Page 5 line 161: please state clearly that all 11 newly sequenced genomes are complete and circularized. If not, please give details.

Table 2: please reduce column 1 and enlarge column 2 to allow using species names as labels (abbreviating genera). In the rest of the manuscript and supplementary material avoid such abbreviations for species, as this makes the text rather difficult to follow.

Figure 3 and Supplementary Figures S4 and later: drawing tRNAs in green makes structures more difficult to read and is not particularly visually pleasing. In addition, check figure resolution, as letters are not showing very clearly in my printout.

Supplementary Table 5: if I understand this correctly, these numbers refer to the spacer right to the gene on the J strand. Please state this in the caption.

Page 13 line 272: nucleotide diversity is generally indicated using the greek character for Pi or the english lowercase italic p. Unless symbol 'Pi' is similarly in general use according to the experience of the Authors, please use a more conventional abbreviation.

Page 15 line 302: 'The 22 Ty. species ... are confirmed as monophyletic' is a wrong statement. A correct rewording for this would be that they 'form a monophyletic group with respect to the two outgroup species'. Not very interesting, as the ingroup is 'assumed' to be monophyletic in all analyses, but at least correct.

Page 15 line 305: Incorrect sentence. Strictly speaking, Zygellini is polyphyletic and Typhlocybini is paraphyletic. An easier way to deal with this would be to define the two groups as mutually paraphyletic.

Page 15 line 316: Incorrect sentence, I think. 'Topologies of the 5 ... tribes' refer to internal relationships between multiple species of each tribe, and I assume this is not what the authors mean. If relationships among tribes is what they mean, I would use 'inter-tribe relationships' or something like this.

Page 15 line 322 and Abstract last line: I do not see any evidence to support 'Erythroneurini as a subtribe of Dikraneurini'. Based on the phylogenetic trees these are two monophyletic groups that cluster as sister groups on the tree. This is totally in line with their being two tribes. The situation of Zygellini and Typlocybini is, on the other hand, less straightforward. If the results of the phylogenetic analysis are taken at face value, they would suggest that the two tribes are invalid, and species may be merged in a single group. Nevertheless, if an established taxonomy is available for the group, I would refrain from suggesting its modification based on a single study. I suggest mentioning the inconsistency and discussing its significance in the light of the existing taxonomy.

Conclusions: please do not repeat information that has been already presented/discussed in the main text. Instead, try to generalize on the overall significance of this study.

Reviewer 2 Report

The manuscript prepared and submitted by Lin and colleagues presents a very interesting and up-to-date work on the phylogenetic relationships between Typhlocybinae based on mitogenomes. Though the work is important, the manuscript has severe flaws that render very difficult for the reader to thoroughly understand its impact.For that I suggest the authors to consult an english speaker and revise the text accordingly, something that will definitely improve the quality and understanding of the text. Below you can find a brief (and not exhaustive) list of issues that should be improved:

Line 9:...evolved group of leafhoppers, with a body length...

Line 46: what do you mean here with "tropical fauna"? please rephrase

Line 49: ...system proposed by...

Line 56: ...act as vectors..

Line 61: What do you mean by "exploited urgently"? Please rephrase

Line 66: What do you mean by the fact that "mitogenome provides genome-wide information"?

Line 73: ...in the past.

Line 80: This is based...

Line 162: ...mitogenomes range from...

Line 171: ...strong AT...

Line 312: As the synonimization of tribes and other taxonomic units is relatively difficult to be supported only by one approach, I would tone down this idea of synonimization between Zygenellini and Typhlocybini.

Line 331: ...with number and order...What do you mean here? Please rephrase otherwise it is difficult to understand.

Line 332: I believe that this sentence regarding the initiation codons, belongs to Results and/or discussion but surely not here.

Lines 338-341: as these Lines are at the very end of your manuscript, I would rephrase them, as it is difficult to get the clear meaning. 

Table 2: I believe that this Table can be transferred to the supplementary material.

Reviewer 3 Report

1. Some misspellings should be checked and corrected.
2. The question which subfamily Typhlocybinae or Deltocephalinae is more diversified seems not to be resolved, depending on arguments used.
3. Put some comments on taxa selected and the reasons for this action, .g. Alebroides group was recently revealed as paraphyletic (Xu et al. 2021). Empoascini seems not to be as uniform as previously believed, similar questions are to be found among Dikraneurioni and Erythroneurini. Several other classification proposals and synonymies were already proposed, accepted or rejected, so the internal classification of Typhlocybine is going to be more messy. Note that Typhlocybini were subdivided into subtribes by Anufriev (1978). For any reliable relationships proposal much more tacxa should be taken into consideration and/or the terminal taxa selected very carefully to cover the diversity and disparity of the microleafhoppers.

Round 2

Reviewer 2 Report

The manuscript has now been significantly improved as the authors have meticulously dealt with the comments and questions raised in the first round of review. For that, I now believe that the manuscript can be accepted for publication in its current firm